# Probing the Inflaton Potential with SKA

Tanmoy Modak[1], Tilman Plehn[1], Lennart Röver[1] and Björn Malte Schäfer[2]

**1** Institut für Theoretische Physik, Universität Heidelberg, Germany
**2** Astronomisches Recheninstitut, Universität Heidelberg, Germany

## Abstract

SKA will be a major step forward not only in astrophysics, but also in precision cosmology. We show how the neutral hydrogen intensity map can be combined with the Planck measurements of the CMB power spectrum, to provide a precision test of the inflaton potential. For a conservative range of redshifts we find that SKA can significantly improve current constraints on the Hubble slow-roll parameters.

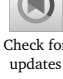
## 1 Introduction

Cosmic inflation [1–3], originally designed as a solution to the flatness and horizon problems, provides a viable mechanism for seeding cosmic structures [4–7]. It can be studied through fluctuations in the cosmic microwave background at early times and through the distribution of dark matter and galaxies in the late Universe. The standard paradigm assumes a spatially homogeneous scalar inflaton field $\varphi$ with a potential $V(\varphi)$ and a coupling to gravity through its energy momentum tensor. If a single inflaton is initialised in a slow-roll state [8–10] it drives an accelerated expansion of the background spacetime. During this exponential expansion

the comoving horizon shrinks and sets the amplitude of fluctuations at the moment of their horizon exit. We can then compute the spectrum of scalar and tensor perturbations from the time evolution of the Hubble function, which in turn is completely determined by the inflaton potential and initial conditions.

Planck's observations of the cosmic microwave background (CMB) temperature and polarisation anisotropies have advanced our understanding of inflation tremendously [11, 12]. The natural next step would be to include observations on a larger range of scales, to further constrain the spectral shape and with that the inflationary potential. Here, structures in the late Universe ideally complement the CMB, but one needs to proceed with caution as non-linear structure formation adds significant amounts of power in the spectra on small scales. In addition, depending on the observational channel, the relationship between the fundamental fields, density or gravitational potential, and the actual observable might be nonlinear, as is the case of the galaxy distribution. In this article we study the distribution of neutral hydrogen, mapped out by the Square Kilometer Array (SKA) [13–24], as a probe of inflation models. The density field on these scales is, to good approximation, in a linear stage of structure formation with Gaussian statistics and can potentially take cosmology to the next level of precision [25–29], even for non perfectly Gaussian fields [30,31], along with a determination of the astrophysical parameters [32]. Combined with with comparatively low systematic influences due to $X$-ray or UV-sources [33–37] or due to baryonic feedback processes [38–40], this should allow us to probe inflationary parameters through 21cm tomography in this window. While we will use idealising assumptions in constraining inflationary parameters in this work, modelling of the reionisation process at high redshift has reached a high degree of sophistication [41–46] and takes care of astrophysical processes, which are likewise modelled in machine learning approaches [47, 48].

Several studies have looked at 21cm neutral hydrogen tomography as a tool for precision measurements of the spectral index $n_s$ and its running [49–55], to constrain the inflationary potential. Focusing on redshift $z = 8 – 10$ we analyze the SKA potential in constraining the Hubble slow-roll (HSR) parameters [56], which are defined either as logarithmic derivatives of the Hubble function. A wide range of scales enters this measurement, ideally from $10^{-2}$ to $1$ Mpc$^{-1}$, and the systematics related to nonlinear structure formation on small scales or nonlinear relation between observable and the density perturbations can be controlled. This will allow SKA to derive tight bounds on the Hubble slow-roll parameters, in combination with the Planck measurements of the CMB spectrum.

In Sec. 2 we discuss the required formalism for inflation assuming a single field $\varphi$ driving the inflation. We validate our approach using the constraints on slow-roll parameters from the Planck 2018 measurements in Sec. 3. We discuss about forecast on the slow-roll parameters from SKA and, combined Planck and SKA in Sec. 4. Finally, we summarize our results.

## 2  Cosmic inflation and cosmic structures

The Planck measurements of the CMB temperature and polarisation anisotropies have been the first systematic probe of inflationary parameters. The spectral index $n_s$, its running $\mathrm{d}n_s/\mathrm{d}\ln k$, and the scalar-to-tensor ratio $r$ have been measured with high precision, and the impact of cosmological parameters and the optical depth $\tau$ has been investigated in detail. Assuming single field inflation, these measurements can be translated into slow-roll parameters, as we will briefly review below.

**Slow-roll inflation**

During cosmic inflation, the evolution of the Universe is dominated by the gravitational effect of a single field $\varphi$, whose energy-momentum content acts as a source of gravity. As $\varphi$ is assumed to conform to the FLRW-symmetries, it can only depend on time. Because its action

$$S = \int d^4x \sqrt{-g} \left( \frac{1}{2} g^{\mu\nu} \nabla_\mu \varphi \nabla_\nu \varphi - V(\varphi) \right) \tag{1}$$

does not contain any dissipative terms or couplings to other fields, it acts as an ideal fluid with density $\rho$, pressure $p$, and a covariantly conserved energy momentum tensor. Variation with respect to $\varphi$ and imposing the FLRW-symmetries gives us the Klein-Gordon equation

$$\ddot{\varphi} + 3\frac{\dot{a}}{a}\dot{\varphi} = -\frac{dV(\varphi)}{d\varphi}. \tag{2}$$

The gravitational effect of $\varphi$ on a FLRW-spacetime with scale factor $a(t)$ is given by the Friedmann equations

$$\left( \frac{\dot{a}}{a} \right)^2 = \frac{8\pi}{3m_{\mathrm{Pl}}^2} \left( \frac{\dot{\varphi}^2}{2} + V(\varphi) \right) \quad \text{and} \quad \frac{\ddot{a}}{a} = -\frac{8\pi}{3m_{\mathrm{Pl}}^2} \left( \frac{\dot{\varphi}^2}{2} - V(\varphi) \right). \tag{3}$$

The limit $\dot{\varphi}^2 \ll 2V(\varphi)$ is referred to as the slow-roll phase. The Klein-Gordon equation together with the first Friedmann equation allow us to write the evolution of the FLRW-universe in terms of the Hubble function $H = \dot{a}/a = d\ln a/dt$,

$$\dot{\varphi} = -\frac{m_{\mathrm{Pl}}^2}{4\pi}H'(\varphi),$$

$$V(\varphi) = -\frac{m_{\mathrm{Pl}}^4}{32\pi^2}[H'(\varphi)]^2 + \frac{3m_{\mathrm{Pl}}^2}{8\pi}H^2(\varphi), \tag{4}$$

with $H' = dH/d\varphi$. In ideal slow roll, the expansion of the Universe is exponential with a constant Hubble function. Deviations are parametrized by

$$\epsilon_H = \frac{m_{\mathrm{Pl}}^2}{4\pi} \left( \frac{H'}{H} \right)^2. \tag{5}$$

which reflects the equation-of-state parameter $w = p/(\rho c^2) \approx -1$, as required by an exponential expansion. Analogously, a small value of

$$\eta_H = \frac{m_{\mathrm{Pl}}^2}{4\pi} \left( \frac{H''}{H} \right) \tag{6}$$

makes sure that slow roll is maintained for a sufficiently long time.

Starting from these two intuitive parameters one defines a full hierarchy of Hubble slow-roll parameters that quantify logarithmic changes to the Hubble function,

$$\lambda_H^{(n)} = \left( \frac{m_{\mathrm{Pl}}^2}{4\pi} \right)^n \left( \frac{(H')^{n-1}}{H^n} \frac{d^{n+1}H}{d\varphi^{n+1}} \right), \qquad n \geq 1, \tag{7}$$

with the usual correspondence $\eta_H = \lambda^{(1)}$, $\xi_H^2 = \lambda^{(2)}$, and $\omega_H^3 = \lambda^{(3)}$. Expanding around the inflaton field value $\varphi_*$ at the horizon crossing with the pivot scale $k_* = 0.05\,\mathrm{Mpc}^{-1}$, the Hubble function can be reconstructed in the observable window defined by the range of observationally accessible spatial scales as

$$H(\varphi) = \sum_{n=0}^{N} \frac{1}{n!} \left. \frac{d^n H}{d\varphi^n} \right|_{\varphi_*} (\varphi - \varphi_*)^n, \tag{8}$$

expressed in terms of the $\lambda_H^{(n)}$.

**Perturbation theory**

A suitable coordinate choice for perturbation theory is comoving gauge, where spatial hypersurfaces are orthogonal to the worldlines of FLRW-observers, and the corresponding gauge invariant quantity is the Mukhanov potential

$$u = a\delta\varphi - \frac{\mathcal{R}}{H}\frac{\partial\varphi}{\partial\eta}. \tag{9}$$

It is constructed from the curvature perturbation $\mathcal{R}$ and the inflationary field perturbation $\delta\varphi$, and $\eta$ is conformal time. Its Fourier modes $u(k)$ evolve according to

$$\frac{\mathrm{d}^2}{\mathrm{d}\eta^2}u(k) + \left[k^2 - \frac{1}{z}\frac{\mathrm{d}^2z}{\mathrm{d}\eta^2}\right]u(k) = 0, \tag{10}$$

where $z = a\dot{\varphi}/H$ and the initial conditions are formally set at

$$u(k, \eta \to -\infty) = \frac{e^{-ik\eta}}{\sqrt{2k}}. \tag{11}$$

This evolution equation is tackled by first solving the background evolution of the FLRW-universe with the Hubble function as a Taylor series and then solving the mode equation. This way we obtain a scale-dependent prediction of $u(k)$ at the end of inflation, where slow roll is violated and the Universe transitions away from exponential expansion. The amplitudes of the Fourier modes define the spectrum of curvature perturbations

$$\mathcal{P}_{\mathcal{R}}(k) = \frac{k^3}{2\pi^2}\left|\frac{u(k)}{z}\right|^2. \tag{12}$$

While perfect slow roll would guarantee scale-independent curvature perturbations and generate a perfect Harrison-Zel'dovich-spectrum, any deviation leads to modulations. They can be computed by mapping the slow-roll parameters onto a logarithmic Taylor expansion of the potential of the type

$$\ln\mathcal{P}_{\mathcal{R}}(k) = \ln A_s + \ln\frac{k}{k_*}\left[(n_s - 1) + \frac{\alpha}{2}\ln\frac{k}{k_*} + \frac{\beta}{3!}\ln^2\frac{k}{k_*} + \dots\right], \tag{13}$$

where the expansion scale $k_*$ is exactly the pivot scale.

For $n_s = 1$ and $\alpha = \beta = 0$ we recover the Harrison-Zel'dovich spectrum $\mathcal{P}_{\mathcal{R}}(k) = \text{const}$. The curvature perturbation spectrum $\mathcal{P}_{\mathcal{R}}(k)$ defined in Eq. (12) serves as an input for the computation of all observables, most notably the CMB temperature and polarisation spectra, as well as for fluctuations in the 21cm brightness.

To determine the constraints on the slow-roll parameters we use the MCMC engine MontePython3 [57, 58], interfaced with the Boltzmann code CLASS III [59, 60] to solve the background and perturbation equations and find the power spectrum. We truncate the series in Eq.(8) at $N = 4$, such that our parameter space is spanned by

$$\left\{\tilde{A}_s, \epsilon_H, \eta_H, \xi_H^2, \omega_H^3\right\}, \tag{14}$$

also denoted as the Hubble slow-roll (HSR) parameters. The parameter $\mathrm{HSR}_0 \equiv \tilde{A}_s$ is defined in Ref. [59]. The primary reason behind this choice of parameterization is to validate our results as well as to compare the potential of SKA to that of Planck 2018 [11]. Note that this parameterization does not depend on the slow-roll approximation. Here, we chose to parameterize $H$ in Eq. (10) as a Taylor expansion with respect to $(\varphi - \varphi^*)$ as given in Eq. (8).

Hence the HSR parameters are not constant in the observable window and evaluated at the pivot scale $k_* = 0.05\ \mathrm{Mpc}^{-1}$ corresponding to the comoving horizon size. However, for the $\tilde{A}_s$, $n_s$, $\alpha$ and $\beta$ parameterization we have truncated the Taylor expansion in Eq. (12) at $\beta$ and assumed the primordial spectrum $\mathcal{P}_{\mathcal{R}}(k)$ is well captured in the observable window of comoving wave numbers.

For the reference cosmological models through out the paper we assume spatially flat $\Lambda$CDM-cosmology, with specific fixed parameters choices

$$
\begin{aligned}
\omega_b &= 2.242 \times 10^{-2}, & \omega_c &= 0.12, \\
\tau_{\mathrm{reio}} &= 0.05678, & h &= 0.6724,
\end{aligned}
\tag{15}
$$

corresponding to our reproduced Planck 2018 measurements [11], as discussed in the next section.

## 3 Planck validation

As illustrated above, the combined system of differential equations for the slow-roll parameters and the mode equation for the amplitudes $u(k)$ predict the spectrum $\mathcal{P}_{\mathcal{R}}(k)$. Any deviation from perfect slow roll induces a scale dependence and a deviation away from the idealised Harrison-Zel'dovich shape. A measurement which is sensitive to slight variations from a pure power law necessarily encompasses a wide range of scales, ideally from the horizon $ck = aH$ to as small scales as possible. Evading nonlinear structure formation on the smallest scales, the requirement of a linear relationship between observable and potential fluctuations conserving all statistical properties and access to a wide range of scales starting at the pivot scale $k_*$ suggests a combination of the CMB at a redshift around $10^3$ and the neutral hydrogen density at a redshift around 10 as a powerful probe of inflationary dynamics.

The established window to inflationary fluctuations are observations of the CMB temperature and polarisation anisotropies [12]. Perturbations on the spectral distribution of photons along a line of sight incorporate baryonic acoustic oscillations and Sachs-Wolfe-type effects and can be cast into the angular spectra $C^{TT}(\ell)$, $C^{EE}(\ell)$ and $C^{TE}(\ell)$. As long as we neglect the mode equation associated with gravitational waves, we set the spectrum of primordial tensor mode to zero and compute the $E$-mode polarisation from the curvature perturbation alone. Combining the three measured spectra to a likelihood with Planck's noise model and a suitable covariance allows us to constrain Hubble slow-role parameters from simulated Planck data and check our results with the conventional $(\alpha, \beta)$-parametrization defined in Eq.(13).

For a first test of our method we fix the background cosmology to a conventional $\Lambda$CDM-model. The noise model uses Gaussian beam shapes and the typical noise levels as specified for Planck. We restrict ourselves up to $N = 4$ in Eq.(8), as done in Ref. [12]. We sample the

Table 1: Mean values and error bars (95% CL) for the slow-roll parameters shown in Fig 1.

| Parameters | mean | 95% CL |
|---|---|---|
| $\tilde{A}_s \times 10^9$ | 2.084 | $[1.978, 2.197]$ |
| $\epsilon_H$ | 0.006095 | $< 0.01518$ |
| $\eta_H$ | $-0.005849$ | $[-0.02804, 0.02104]$ |
| $\xi_H^2$ | 0.01133 | $[-0.1498, 0.1797]$ |
| $\omega_H^3$ | 0.5182 | $[-1.213, 2.309]$ |

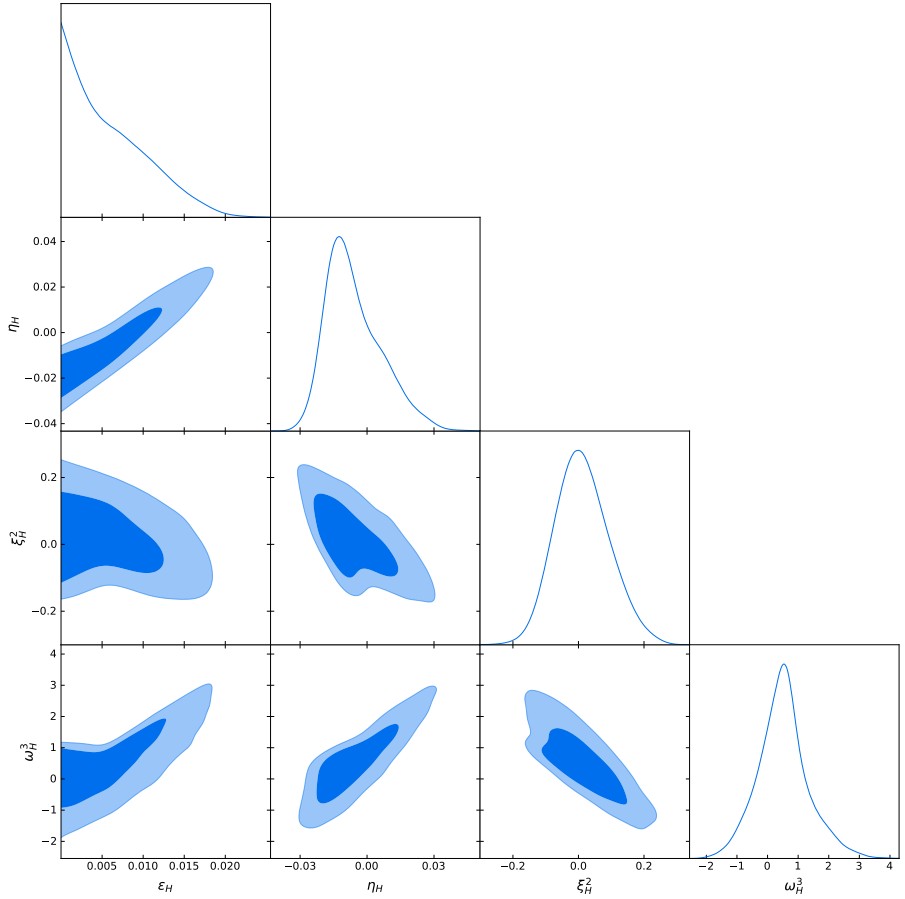

Figure 1: Marginalized joint distributions for parameter pairs at 68% and 95% confidence and marginalized distributions for individual parameters from the 5-dimensional likelihood of the slow-roll parameters in Eq.(14), also marginalized over the cosmological parameters in Eq.(15). We use the joint TT,TE,EE+lowE Planck data, these results should be compared with Fig. 13 of Ref. [12].

slow-roll parameters with flat priors [12,61,62]. The reconstructed inflation parameters from TT,TE,EE+lowE data are shown in Fig. 1 and in Tab. 1. Our results are in good agreement with the dashed contours of Fig. 13 of Planck 2018 [12]. Our values shown in Tab. 1 are mildly weaker than the values shown in Tab. 7 of Ref. [12], as we to not include BK15 and lensing data. For illustration, we also show the angular power spectra for the TT and EE correlations with the respective noise for few representative samples from our parameter scan in Fig. 2.

While the primary aim of this paper is to estimate the potential of SKA and 21cm tomography in measuring the inflaton potential, we need to keep in mind that any SKA measurement will be combined with the Planck CMB constraints. This means we first need to understand the way this correlated set of fundamental parameters affects the CMB power spectra. To illustrate the relation between the different model parameters, we start with a set of ten 2-dimensional parameter scans, fixing three parameters of the 5-dimensional model space defined in Eq.(14). For each 2-dimensional scan we set the remaining three parameters to the mean values given in Tab. 1. We can then assume that the maximum in the 2-dimensional scan should also reproduce the mean values in Tab. 1, but with a correlated uncertainty. In Fig. 3 we show these 2-dimensional parameter planes and confirm that for the combined Planck measurements there do not exist especially strong correlations. In Tab. 2 we give the mean values and the 95% confidence level limits for the 2-dimensional parameter planes shown in Fig. 3.

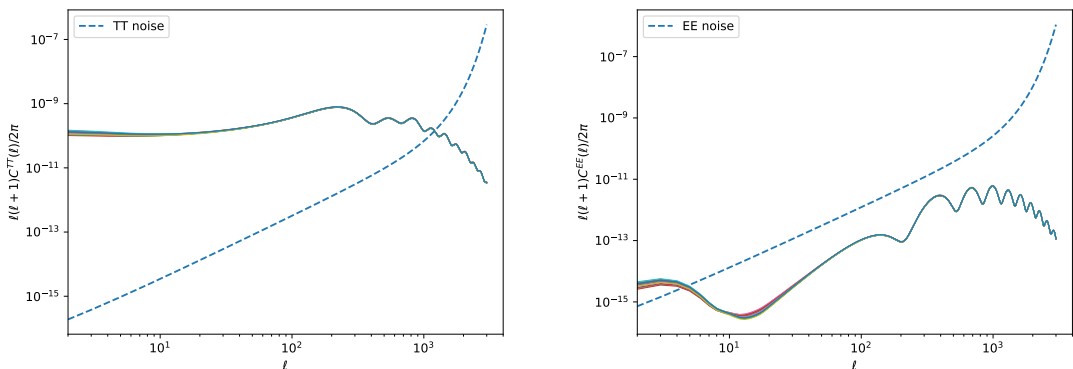

Figure 2: Angular power spectra for the TT (left) and EE (right) correlations for representative samples of the slow-roll parameters along with respective noise power spectrum.

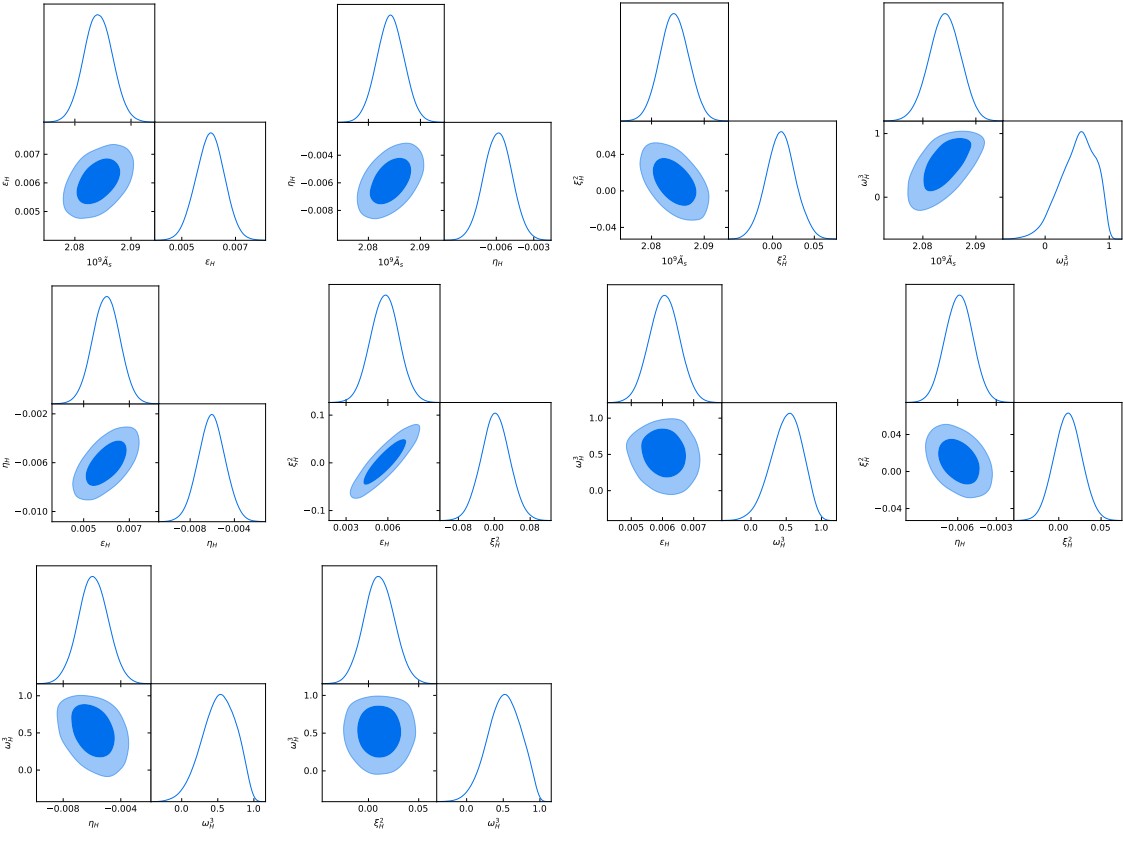

Figure 3: Sliced 2-dimensional likelihoods for the slow-roll parameters from CMB data.

Table 2: Mean values and error bars for 2-dimensional contours of the slow-roll parameters space for the CMB (Fig. 3), 21cm hydrogen spectrum (Fig. 4) and their combination (Fig. 8).

| Parameter | Planck | | SKA | | SKA+Planck | |
|---|---|---|---|---|---|---|
| | mean | 95% CL | mean | 95% CL | mean | 95% CL |
| $\tilde{A}_s \times 10^9$ vs $\epsilon_H$ | 2.0842 | [2.0792, 2.0892] | 2.08401 | [2.08314, 2.08489] | 2.08403 | [2.08318, 2.08491] |
| | 0.0061 | [0.0050, 0.0071] | 0.006096 | [0.006050, 0.006141] | 0.006096 | [0.006052, 0.006142] |
| $\tilde{A}_s \times 10^9$ vs $\eta_H$ | 2.0842 | [2.0791, 2.0894] | 2.08400 | [2.08331, 2.08469] | 2.08401 | [2.08333, 2.08468] |
| | $-0.0059$ | [$-0.0081, -0.0037$] | $-0.005849$ | [$-0.005941, -0.005756$] | $-0.005850$ | [$-0.005940, -0.005758$] |
| $\tilde{A}_s \times 10^9$ vs $\xi_H^2$ | 2.0843 | [2.0791, 2.0897] | 2.08400 | [2.08365, 2.08436] | 2.08400 | [2.08365, 2.08436] |
| | 0.010 | [$-0.024, 0.044$] | 0.01134 | [0.01052, 0.01218] | 0.01134 | [0.01055, 0.01215] |
| $\tilde{A}_s \times 10^9$ vs $\omega_H^3$ | 2.0841 | [2.0782, 2.0898] | 2.08400 | [2.08379, 2.08420] | 2.08400 | [2.08381, 2.08420] |
| | 0.51 | [$-0.02, 0.97$] | 0.518 | [0.498, 0.537] | 0.518 | [0.498, 0.537] |
| $\epsilon_H$ vs $\eta_H$ | 0.0060 | [0.0048, 0.0072] | 0.00609 | [0.00593, 0.00625] | 0.00609 | [0.00593, 0.00624] |
| | $-0.0061$ | [$-0.0085, -0.0037$] | $-0.00586$ | [$-0.00627, -0.00545$] | $-0.00587$ | [$-0.00628, -0.00549$] |
| $\epsilon_H$ vs $\xi_H^2$ | 0.0058 | [0.0038, 0.0078] | 0.006095 | [0.006066, 0.006123] | 0.006095 | [0.006066, 0.006123] |
| | 0.002 | [$-0.059, 0.065$] | 0.0113 | [0.0101, 0.0126] | 0.0113 | [0.0101, 0.0126] |
| $\epsilon_H$ vs $\omega_H^3$ | 0.00606 | [0.00513, 0.00699] | 0.006095 | [0.006085, 0.006105] | 0.006095 | [0.006085, 0.006105] |
| | 0.51 | [0.07, 0.92] | 0.518 | [0.499, 0.536] | 0.517 | [0.498, 0.536] |
| $\eta_H$ vs $\xi_H^2$ | $-0.0059$ | [$-0.0080, -0.0038$] | $-0.005848$ | [$-0.005933, -0.005762$] | $-0.005848$ | [$-0.005935, -0.005760$] |
| | 0.011 | [$-0.020, 0.043$] | 0.0114 | [0.0099, 0.0128] | 0.0113 | [0.0099, 0.0128] |
| $\eta_H$ vs $\omega_H^3$ | $-0.0059$ | [$-0.0079, -0.0039$] | $-0.005849$ | [$-0.005875, -0.005822$] | $-0.005849$ | [$-0.005875, -0.005822$] |
| | 0.52 | [0.07, 0.94] | 0.518 | [0.498, 0.537] | 0.517 | [0.498, 0.536] |
| $\xi_H^2$ vs $\omega_H^3$ | 0.011 | [$-0.018, 0.040$] | 0.01133 | [0.01083, 0.01183] | 0.01132 | [0.01082, 0.01183] |
| | 0.51 | [0.09, 0.93] | 0.518 | [0.497, 0.538] | 0.517 | [0.496, 0.538] |

## 4  SKA projections

The second window to the matter spectrum at relatively high redshift are intensity fluctuations of the 21cm hydrogen line. We focus on the redshifts range between 8 and 10. The upper bound avoids the position dependence of the spin temperature, since the spin temperature couples to the gas temperature through the Wouthuysen-Field effect in this redshift range [49]. The lower bound allows us to avoid position-dependent reionization, as there is still nearly no reionized helium. Since this redshift regime probes patterns from before the reionization started, the neutral hydrogen fraction is $\overline{x}_H = 1$ and we can identify the power spectrum of the neutral hydrogen perturbations $P_{\mathrm{HI}}(k)$ with the matter power spectrum $P_\delta(k, z)$. This means the two-point temperature correlations of the 21cm intensity can be expressed as [53, 63]

$$\langle \Delta T_{21}(\mathbf{k}) \Delta T_{21}(\mathbf{k}') \rangle \equiv P_{21}(\mathbf{k}, z)(2\pi)^3 \delta(\mathbf{k} - \mathbf{k}'), \tag{16}$$

where $\Delta T_{21}(\mathbf{k})$ is the Fourier transformation of the difference between the 21cm temperature $T_{21}(\mathbf{x})$ with,

$$P_{21}(\mathbf{k}) = \left[ \mathcal{A}(z) + \overline{T}_{21}(z)\mu^2 \right]^2 P_{\mathrm{HI}}(k, z). \tag{17}$$

Here the parameter $\mu \equiv k_\parallel/k$ is the cosine between the line of sight $k_\parallel$ and the absolute value $k$ and the $\overline{T}_{21}(z)$ is the average 21cm temperature at redshift $z$ where the function $\mathcal{A}(z)$ can be found from Refs. [63, 64]. The $P_{\mathrm{HI}}(k, z)$ is the spectrum of the neutral hydrogen density fluctuation which we assumed to be equal to the matter spectrum $P_\delta(k, z)$, implying zero (re)ionisation [65]. Before the beginning of reionization the function $\mathcal{A}(z)$ and the average temperature at a specific redshift can be approximated as [53]

$$\mathcal{A}(z) = \overline{T}_{21}(z) = 27.3 \text{ mK} \times \overline{x}_H \frac{T_s - T_\gamma}{T_s} \left( \frac{1+z}{10} \right)^{1/2}. \tag{18}$$

During the epoch of recombination the spin temperature can be taken to be much larger than the photon temperature due to the Wouthuysen-Field effect. The gas temperature in the intergalactic medium is heated by $X$-ray photons up to hundreds of Kelvin [49]. This allows us to drop the temperature factor, which reduces the previous expression to

$$\mathcal{A}(z) = \overline{T}_{21}(z) = 27.3 \text{ mK} \times \overline{x}_{\text{H}} \left( \frac{1+z}{10} \right)^{1/2}. \tag{19}$$

This way, the 21cm-intensity and the matter distribution are linked in the most straightforward way possible, with a uniform modelling of the relationship between fundamental field and observable [66, 67], ignoring cross-correlations [68, 69] and taking into account velocities only [70], while ignoring structures beyond that of a continuous Gaussian random field such as halo formation [71].

The instrumental noise power spectrum in Fourier space can be expressed as [72, 73]

$$P_{21}^N = \frac{\pi T_{\text{sys}}^2}{t_o f_{\text{cover}}^2} d_A^2(z) y_\nu(z) \frac{\lambda^2(z)}{D_{\text{base}}^2}, \tag{20}$$

where $D_{\text{base}}$ is the baseline of the antenna array that is uniformly covered up to a fraction $f_{\text{cover}}$ and $t_o$ is the observation time, with $\lambda(z)$ the 21cm-transition wavelength at redshift $z$ (for optimisations of the design, please refer to [74]). The conversion function from frequency $\nu$ to line of sight $k_\parallel$ is $y_\nu = 18.5((1+z)/10)$ Mpc/MHz, while the system temperature can be parameterized as [53]

$$T_{\text{sys}} = 180 \text{ K} \times \left( \frac{\nu}{180 \text{ MHz}} \right)^{-2.6}. \tag{21}$$

Here the frequency is the 21cm transition at redshift $z$, $\nu = \nu_0/(1+z)$. We take the observation time as $t_o = 10000$ hours (hrs) for our analysis, however we also provide results for 1000 hrs for comparison. The baseline $D_{\text{base}} = 1$ km is taken to be the baseline specified for SKA-LOW in Ref. [22]. The coverage fraction in the nucleus of the antenna array can be computed as [72]

$$f_{\text{cover}} = N_a \frac{D^2}{D_{\text{base}}^2}, \tag{22}$$

where $N_a$ is the number of antennas while $D$ is their diameter. For SKA-LOW [22] the coverage fraction is approximately $f_{\text{cover}} \approx 0.0091$.

For a specific redshift bin centered at $z_i$ the $\chi^2$ functional can be expressed as [53, 75]

$$\chi_i^2 = \frac{f_{\text{sky}}}{2} \frac{\text{Vol}_i}{(2\pi)^3} \int_{k_{\text{min}}}^{k_{\text{max}}} dk (2\pi k^2) \int_{-1}^1 d\mu \frac{[P_{21}(\mathbf{k}, z, \theta) - P_{21}^{fid}(\mathbf{k}, z, \theta_{\text{fid}})]^2}{[P_{21}(\mathbf{k}, z, \theta) + P_{21}^N(z)]^2}, \tag{23}$$

where subscripts $i$ denote the redshift bin and $\theta = \{\tilde{A}_s, \epsilon, \eta, \xi, \omega\}$. The comoving volume of the redshift bin $\text{Vol}_i$ can be computed as a spherical shell in comoving distance $r(z_i)$ and $r(z_{i-1})$, where $z_i$ and $z_{i-1}$ are the edges of the redshift bin of interest. The expression to compute the volume reads approximately as

$$\text{Vol}_i = \frac{4}{3} \pi \left( r(z_i)^3 - r(z_{i-1})^3 \right), \tag{24}$$

which is over the redshift range considered very accurate in comparison to integration over the volume evolution, due to the fine slicing in redshift.

For our analysis we take the 22 equally spaced redshift bins in the region $z \in [8, 10]$. The comoving wave numbers are bounded from above by the non linear scale which we set as

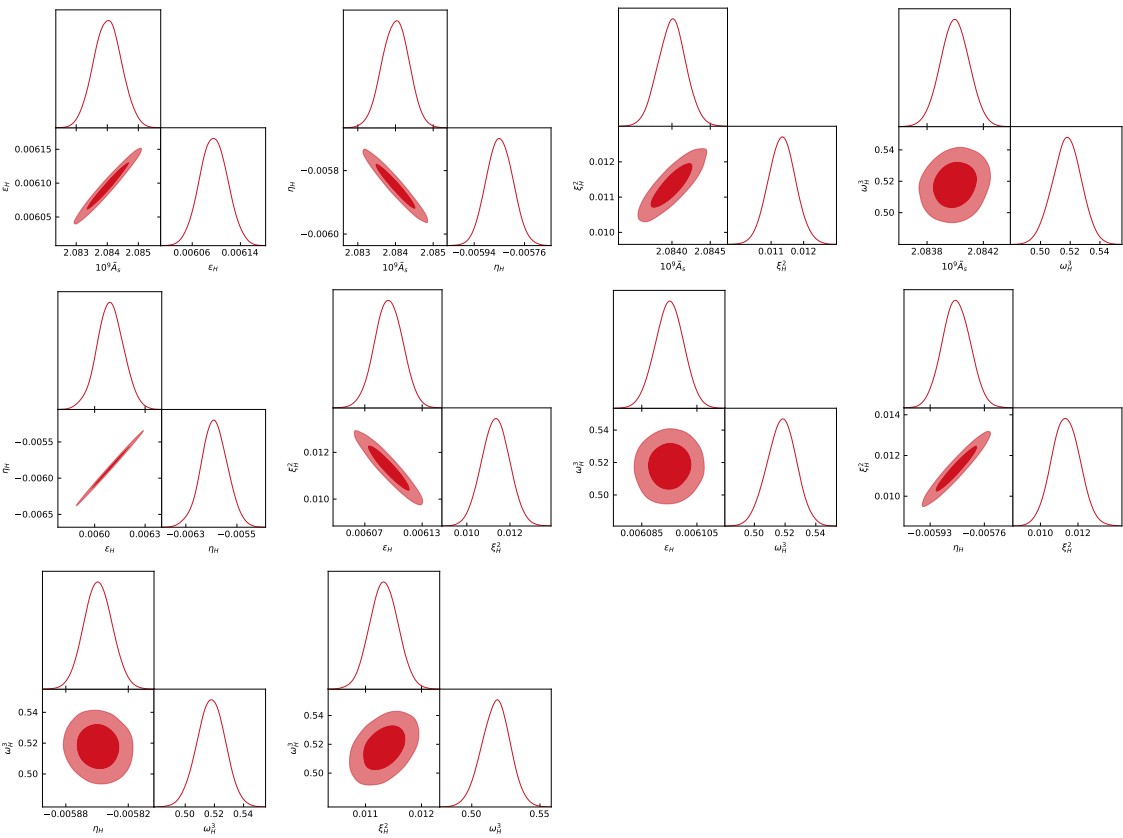

Figure 4: Sliced 2-dimensional likelihoods for the slow-roll parameters from SKA projections.

$k_{\mathrm{NL}} = 1\mathrm{Mpc}^{-1}$, as in Ref. [53]. On the other hand, the astrophysical foregrounds will cut off wave numbers smaller than $k^{\min} \approx 10^{-2}$ Mpc$^{-1}$ [53]. Summing up all the different $\chi_i^2$ we get the overall $\chi^2 = \sum_i \chi_i^2$. The fiducial power spectrum $P_{21}^{\mathrm{fid}}(\mathbf{k}, z, \theta_{\mathrm{fid}})$ is computed according to Eq.(17) and $f_{\mathrm{sky}}$ is set to 0.58 according to Ref. [76]. The parameters $\theta$ are used to compute the matter power spectrum. For the computation of the fiducial power spectrum we choose the mean values for these parameters based on the Planck likelihoods.

Whenever we combine the data sets, we assume that CMB and 21cm data are uncorrelated. This assumption could be challenged if one takes into account effects such as gravitational lensing on the radiation backgrounds by the same structures or correlated secondary anisotropies [77, 78]. We ignore such effects also because they are expected to remain subleading compared to the primary fluctuations of the two radiation backgrounds.

**Slow-roll parameters from SKA**

In this section we discuss the potential of 21cm tomography in constraining the slow-roll parameters in detail. As we shall see shortly a clear hierarchy in sensitivity of observables on cosmological parameters, we simply keep the background cosmology fixed to the $\Lambda$CDM given in Eq.(15). With this vanilla parameter choice, the SKA data only constrains the inflationary potential through the slow-roll parameters defined in Eq.(14).

To understand the correlations between different slow-roll parameters we first plot all possible 2-dimensional Markov chains in Fig. 4 based on the likelihood discussed in Sec. 4. The figure should be compared with Fig. 3 where we show the 2-dimensional Markov chains for Planck 2018 data. The corresponding mean values and errors are presented in Tab. 2

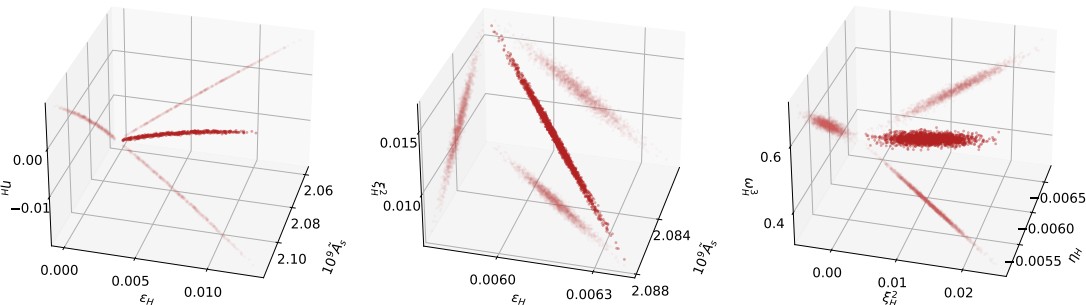

Figure 5: Sliced 3-dimensional likelihood ellipsoids for a selection of slow-roll parameters from SKA projections.

which can be compared with similar one for Planck 2018 as given in Tab. 2. It is clear that SKA offers more stringent constraints on slow-roll parameters then Planck for all 2-dimensional combinations of slow-roll parameters. As for Fig. 3 we set the all remaining parameters of our 2D analyses to the mean values given in the Tab. 1.

In a similar fashion we can study 3-dimensional Markov chains for the slow-roll parameters, illustrating a few combinations in Fig. 5. In Fig. 6 we project the 3-dimensional error ellipses into two dimensions and superimpose the 2-dimensional chains from Fig. 4. We also plot 2D slices from the 3-dimensional Markov chains and superposed them with 2D Markov chains of Figs. 11 and 12 for all slow-roll parameter combinations in App. A. These figures illustrates that the 3-dimensional Markov chains still reflect the correlations observed in 2D chains. For the full set of slow-roll parameters from Eq.(14) we rely on the combination with the Planck data, where the numerics are less challenging than for SKA alone.

The observed improvement in sensitivity on slow-roll parameters provided by SKA originates from the wide range of scales that are probed and that the range of accessible scales stretches to small spatial scales, too, giving SKA in comparison to Planck a better lever to constrain the effect of slow-roll parameters on the spectrum. In Fig. 7 we illustrate variations in the spectrum for a representative selection of samples of the slow-roll parameters. We show the largest scales close to the pivot-scale, as probed by Planck, and the smallest scales, where SKA plays out its unique sensitivity and resolution. Comparing Fig. 7 and Fig. 2 one should keep in mind that on the largest scales there is a significant cosmic variance, which is not included in the shown noise levels, such that the constraining power of CMB-spectra on the largest scales remains limited. We have chosen $h$ to be consistent with the comparatively low values from the CMB in all our forecasts, as both observations probe the high-redshift universe. The choice of a higher value of $h$ as reported from low-redshift observations would not have a strong influence on the constraints of slow-roll parameters, as preliminary tests suggest.

**Slow-roll parameters from SKA and Planck**

To illustrate how the combined SKA and Planck likelihoods constrain the slow-roll parameters, we again start with the 2-dimensional contours in Fig. 8. The corresponding mean values and 95% CL limits are included in Tab. 2. The projected 2-dimensional constraints from the Planck and SKA combination are extremely similar to the SKA limits alone, as expected from the weaker Planck limits shown in Fig. 3. This is expected since the constraints from SKA in the 2D parameters sets are much stronger than the Planck as can be compared from Tab. 2. We provide the 3-dimensional constraints from the Planck and SKA combination in Fig. 13 in the Appendix. Unlike SKA likelihood alone as in previous section we find that 5-dimensional slow-roll parameters converge well, since the Planck likelihood cuts off approximately flat

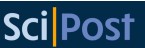

Figure 6: Marginalized 3-dimensional likelihoods from Fig. 5, compared with the 2-dimensional likelihoods in Fig. 4.

directions. The marginalized 2D contours from the 5-dimensional Markov chains for the slow-roll parameters given in Eq.(14) are shown in Fig. 9.

We finally provide constraints on the full set of slow-roll parameters from Planck alone and from Planck and SKA combined in Fig. 10, also varying cosmological parameters $\omega_b$, $\omega_c$, $\tau_{\text{reio}}$ and $h$. These limits can be compared directly to the final Planck results reproduced in Fig. 1 and shown as dashed contours. We assume total observation times of 1000 and 10000 hours for

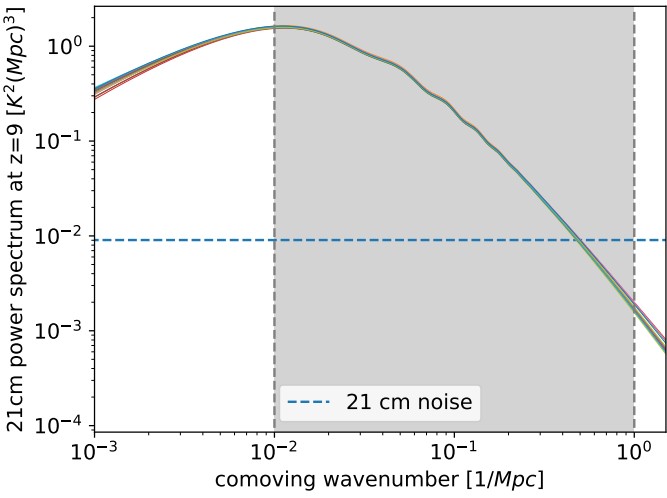

Figure 7: Comparison between 21cm power spectrum and the noise power spectrum computed through Eq.(20). The gray lines denote the maximal and minimal scales considered.

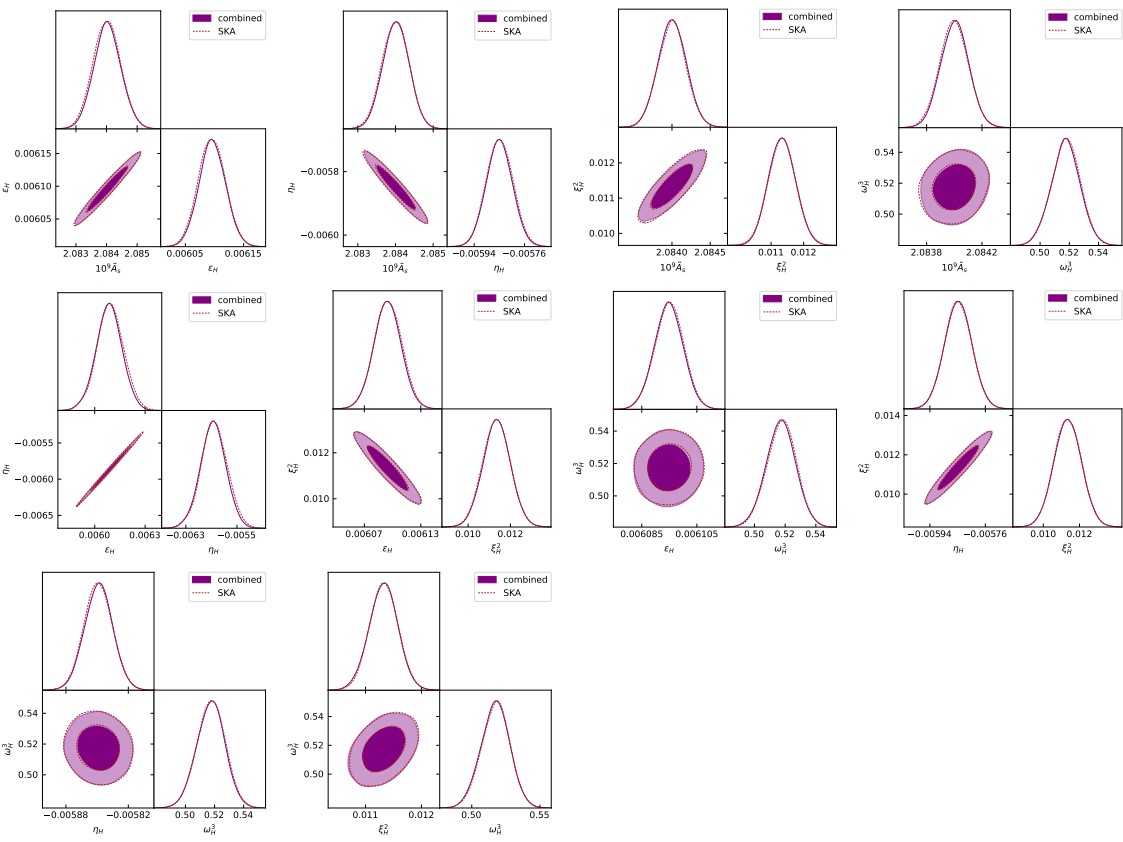

Figure 8: Sliced 2-dimensional likelihoods for the slow-roll parameters from a combination of Planck and SKA projections. We also show the SKA-only contours from Fig. 4 as red dashed lines, the results should be compared to the Planck results shown in Fig. 3.

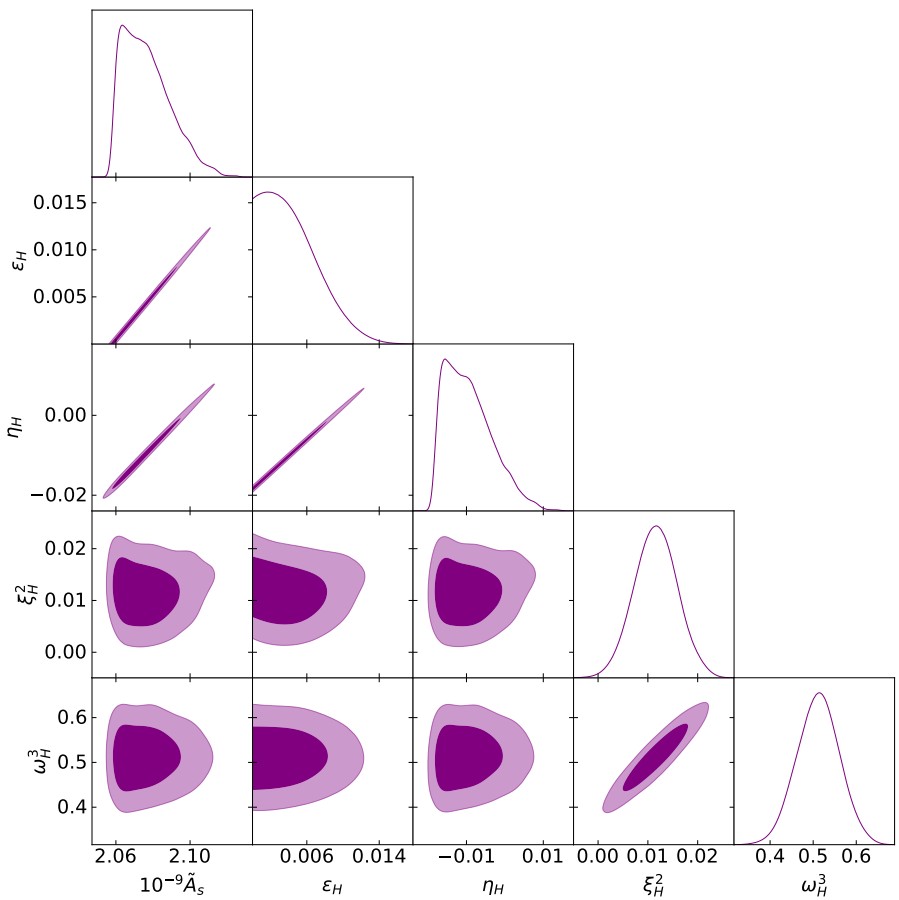

Figure 9: Marginalized 68% and 95% CL contours from the 5-dimensional likelihood of the slow-roll parameters in Eq.(14). We combine the 2018 Planck results with SKA projections assuming 10000 hrs observation time.

SKA. The corresponding mean and 95% CL limits are summarized in Tab. 3. It is clear that the constraints from the combined likelihoods are much stronger than the Planck 2018 data alone. In particular, we find the combined constraints are about one order of magnitude stronger than the Planck constraints for the slow-roll parameters $\xi_H^2$ and $\omega_H^3$. Moreover, the combined data can constrain the slow-roll parameters $\xi_H^2$ and $\omega_H^3$ more stringently than that of $\epsilon_H$ and $\eta_H$, as can be easily seen from Fig. 10 and Tab. 3. The sensitivity gain for the higher-order slow-roll parameters $\xi^2$ and $\omega^3$ is related to the fact that those parameters impact the shape of the

Table 3: Mean values and 95% CL error bars for the slow-roll parameters from Planck 2018 data and combined Planck plus SKA with 1000 hrs or and 10000 hrs observation time, marginalized over cosmological paramers, and corresponding to Fig. 10.

| Parameter | Planck | | SKA+Planck (1000 hrs) | | SKA+Planck (10000 hrs) | |
|---|---|---|---|---|---|---|
| | mean | 95% CL | mean | 95% CL | mean | 95% CL |
| $\tilde{A}_s \times 10^9$ | 2.084 | $[1.978, 2.197]$ | 2.075 | $[2.046, 2.110]$ | 2.075 | $[2.048, 2.106]$ |
| $\epsilon_H$ | 0.006095 | $< 0.01518$ | 0.0043 | $< 0.0101$ | 0.0041 | $< 0.00951$ |
| $\eta_H$ | $-0.005849$ | $[-0.02804, 0.02104]$ | $-0.0097$ | $[-0.0198, 0.0032]$ | $-0.0101$ | $[-0.0197, 0.0022]$ |
| $\xi_H^2$ | 0.01133 | $[-0.1498, 0.1797]$ | 0.011 | $[-0.002, 0.024]$ | 0.011 | $[0.000, 0.023]$ |
| $\omega_H^3$ | 0.5182 | $[-1.213, 2.309]$ | 0.51 | $[0.33, 0.67]$ | 0.51 | $[0.41, 0.61]$ |

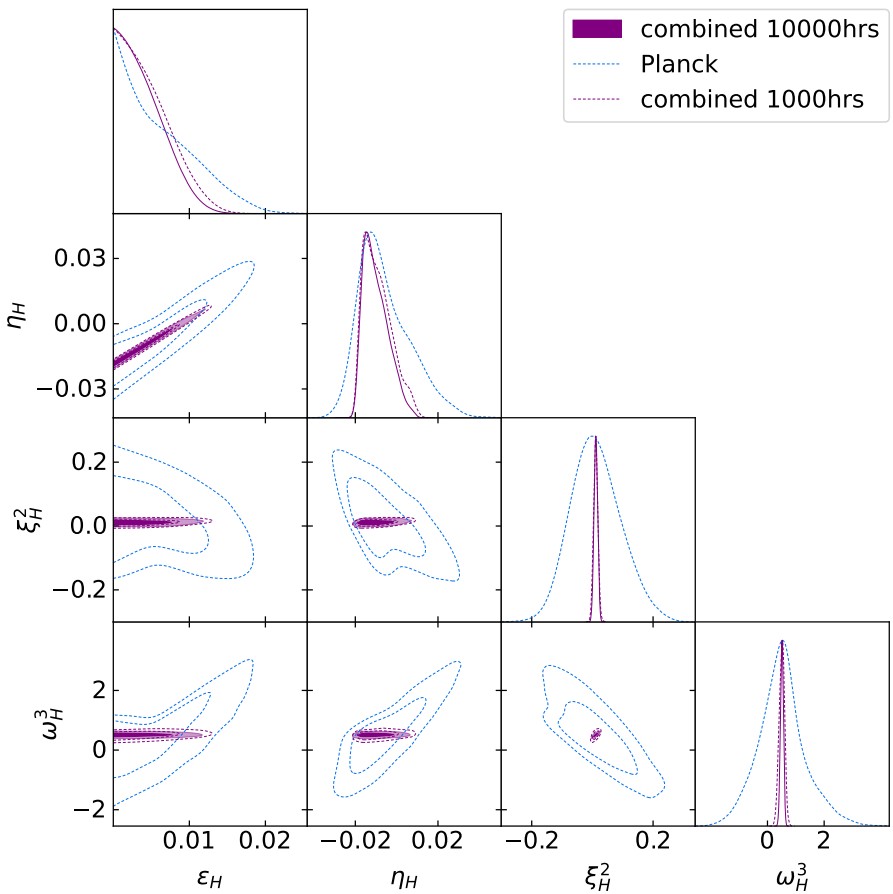

Figure 10: Marginalized 68% and 95% CL contours from the 5-dimensional likelihood of the slow-roll parameters in Eq.(14), also marginalized over the cosmological parameters in Eq.(15). We combine the 2018 Planck results with SKA projections assuming 10000 hrs (purple solid) and 1000 hrs (purple dashed) observation time and overlay the Planck contours from Fig. 1

spectrum strongest far away from the pivot scale $k_*$, similar to the parametrization given in Ref. 13. As the pivot scale $k_*$ is chosen to be the largest scale in the problem and covered by CMB-observations (albeit with a limitation due to cosmic variance), significant improvement is provided by the smallest scales, far away from $k_*$.

A non-Harrison-Zel'dovich form of the spectrum in terms of $\alpha$ and $\beta$ can be measured with CMB-S4 experiments, as well as precision 21cm-surveys [53]. To allow for a comparison we present our results in terms of these parameters in Fig. 14 and Tab. 4 of the Appendix. The mapping to slow-roll parameters is not unambiguous because of its nonlinearity, so we prefer to work with the slow-roll parameters directly.

# 5 Outlook

We have presented constraints on the single-field inflationary potential in terms of the Hubble slow-roll parameters from the CMB temperature and polarisation spectra combined with the 21cm brightness fluctuations. We compute the spectrum of curvature perturbations for a sample of initial values, parameterizing the Hubble function in terms of the scalar field amplitude in a truncated Taylor-expansion. The field itself, the background cosmology and the

mode equations for perturbations are evolved together to yield a curvature perturbation spectrum at horizon-exit. We then evolve the modes using a Boltzmann-code and link them to CMB temperature and polarisation anisotropies, as well as the matter power spectrum at low redshift, from which we model the fluctuations in the 21cm spectrum. All spectra with their instrumental noise levels and covariances form a likelihood for the slow-roll parameters, parameters of the $\Lambda$CDM background cosmology, and parameters inherent to the observational channels such as the optical depth.

Violation of slow roll causes scale-dependent variations of the scale-invariant Harrison-Zel'dovich spectrum. It can be described by a Taylor-expansion of the curvature perturbation spectrum in terms of logarithmic wave number, relative to a pivot-scale close to the horizon. Both, primary CMB spectra and 21cm intensity fluctuations probe a wide range of scales with a linear relation between observable and fundamental field. This range is key to the sensitivity to the inflationary potential, as the variation of the shape of the spectrum with logarithmic wave number is generically small. In terms of the Hubble slow-roll parameters especially the SKA limits showed strong degeneracies and increasingly loose bounds on higher-order parameters. We recovered a hierarchy in precision, where $\epsilon_H$ and $\eta_H$ are measured at a level of $\sim 10^{-2}$, followed by $\xi_H^2$ at $10^{-1}$ and $\omega_H^3$ just slightly better than order-one. The improvement of SKA over Planck, in particular on $\xi_H^2$ and $\omega_H^3$, is driven by small scales, where deviations from the Harrison-Zel'dovich shape far away from the pivot scale $k_*$ become important.

Naturally, one would like to extend the scale range to higher wave numbers and include low-redshift probes of the cosmic large-scale structure such as weak cosmic shear or galaxy clustering at redshifts around unity. However, on such scales nonlinear structure formation starts to dominate. Similarly, small scales at higher redshift can be probed by Lyman-$\alpha$ measurements, which requires a detailed understanding of baryonic dynamics. Additional constraints on the spectral shape on small scales will eventually come from limits on primordial black holes. We leave these additional handles for future analyses and instead follow a very conservative approach. Even with a limited range of scales and a narrow redshift window we confirm that SKA will provide excellent limits on the inflationary potential, pushing precision cosmology significantly beyond the CMB measurements by Planck.

## Acknowledgments

We thank Robert F. Reischke and Michel Luchmann for discussions. We also thank Julien Lesgourgues for communication. TM is supported by Postdoctoral Research Fellowship from Alexander von Humboldt Foundation. The research of TP is supported by the Deutsche Forschungsgemeinschaft (DFG, German Research Foundation) under grant 396021762 – TRR 257 *Particle Physics Phenomenology after the Higgs Discovery*. This work was supported by the Deutsche Forschungsgemeinschaft (DFG, German Research Foundation) under Germany's Excellence Strategy EXC 2181/1 - 390900948 (the Heidelberg STRUCTURES Excellence Cluster).

## A   Appendix

### 2D slices from 3D Markov Chains

The 3-dimensional Markov chains generated with the SKA likelihoods in 4 are consistent with the two dimensional Markov chains. Fig. 11 and 12 show the comparison between fraction of the three dimensional Markov chains and the two dimensional Markov chains. To select

appropriate points from the three dimensional Markov chains one parameter is chosen and only those points within a small region around its mean value found in 1 are taken into account to generate the figures. For each three dimensional Markov chain three of these sliced chains are generated. The cut chains exhibit similar mean parameter values and contours as the two dimensional Markov chains.

Adding the Planck likelihoods to the 3-dimensional parameter estimation yields results very similar to the 3-dimensional SKA limits alone. The marginalized likelihoods are shown in Fig. 13. When constraining $\tilde{A}_s$, $\epsilon_H$ and $\eta_H$ at the same time the marginalized distributions for each of the parameters become less wide. The Planck likelihood constrains the very edges of the strongly correlated parameters, allowing for an easier numerical evaluation.

### SKA with spectral index and running

The primordial power spectra $\mathcal{P}_{\mathcal{R}}(k)$ can also be parameterized by the spectral index $n_s$ and its running $\alpha$ and $\beta$, which can also be constrained by 21cm cosmology. We also provide the constraints on these parameters from SKA alone in Fig. 14 for both 10000 (red solid) and 1000 hrs (red dashed) for comparison. The corresponding mean values and 68% error bars are given in Tab. 4. We find that our constraints are a bit stronger than that found by Ref [53] for 1000 hrs observation time.

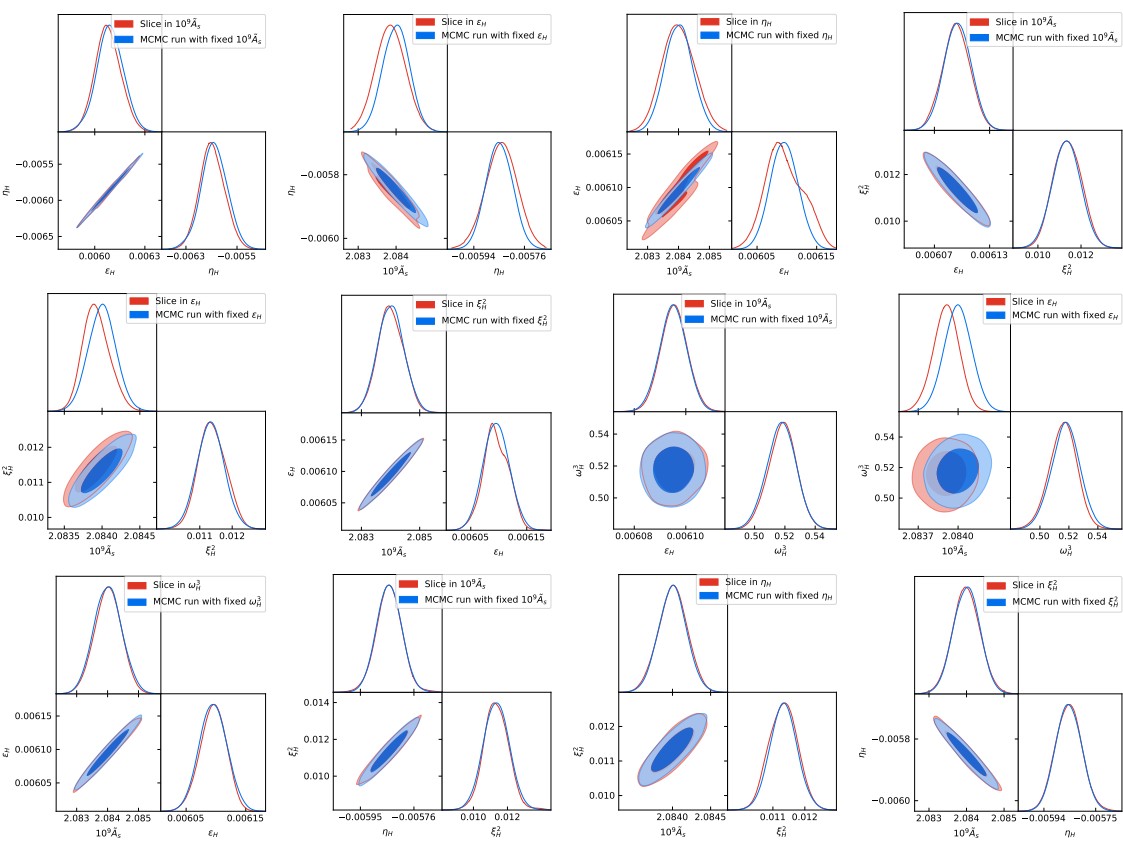

Figure 11: Comparison between slices and 2D contours using the SKA projections. Slices are taken from the 3D chains in Fig. 6.

Figure 12: Comparison between slices and 2D contours using the SKA projections. Slices are taken from the 3D chains in Fig. 6.

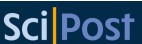

Figure 13: Sliced 3-dimensional likelihoods for the slow-roll parameters from a combination of Planck and SKA projections.

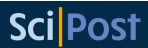

Figure 14: The contours for SKA likelihoods (red) and overlaid SKA with lower observation time of 1000 hours (red dashed lines).

Table 4: Best fit parameter values and error bars for SKA likelihoods constructed as in 4 for primordial spectrum computation with spectral index and its runnings.

| Parameter | mean | 68% CL 10000 hrs | 68% CL 1000 hrs |
|---|---|---|---|
| $10^{10}A_s$ | 21.157 | ±0.064 | ±0.092 |
| $n_s$ | 0.9646 | ±0.0031 | ±0.0043 |
| $\alpha$ | −0.0080 | ±0.0018 | ±0.0029 |
| $\beta$ | 0.00710 | ±0.00083 | ±0.0016 |

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
