# Peer review of "Probing the Inflaton Potential with SKA"

_SciPost Physics, doi:SciPost Phys. Core 5, 037 (2022)_

## Round 1 · Referee Report · Anonymous (Referee 1) · 2022-2-16

Strengths

The author forecasted errors of general parameters appeared in inflation models, using future observations by SKA.

Weaknesses

The way of their presentation would be highly unclear.

Report

By adopting specifications of SKA, the authors forecasted possible future constraints on parameters in inflaton's potential. The author's attempt is very meaningful. However, I still have some questions as follows.

1) The authors used the existing constraints on the parameters obtained from Planck as a prior. However, we know there are some uncertainties even on choices for the values of the constraints. If the authors adopted the most milder bounds on them, how do the results change?

2) About uncertainties coming from the H0 tension, how did the authors resolve the problem in their choices for actual values of the free parameters?

3) Even for the ionization fraction of electron, chi, there should exists the corresponding spatial fluctuation. How did the author treat the chi's fluctuation in their analysis?

4) It seems that the authors assumed the constant slowroll parameters (or constant spectral index, the running, and the running-of-running, ...) as a function of the wave number (k). I think this is not realized in a concrete inflation model for relevant ranges of k observed by the SKA. I encourage the authors to discuss the validity for their methods in the text.

Requested changes

I encourage the authors to answer my questions written in my report.

  • validity: ok
  • significance: ok
  • originality: ok
  • clarity: low
  • formatting: good
  • grammar: reasonable

Author:  Lennart Röver  on 2022-05-03  [id 2439]

(in reply to Report 1 on 2022-02-16)

The reply to the referee is attached as a pdf file.

We have also resubmitted v2 of the paper through the resubmission option.

Attachment:

referee_reply.pdf

Anonymous on 2022-05-04  [id 2441]

(in reply to Lennart Röver on 2022-05-03 [id 2439])

The authors answered all of my questions. I recommend this paper to be published in SciPost Physics.

---

## Round 2 · Author Response

We have added a reply on the submission page as an attachment.

---

## Round 2 · List of Changes

Changes in the manuscript are written in red.

---

## Editorial Decision

published